# Salivary Metabolomics as a Diagnostic Tool: Distinct Metabolic Profiles Across Orofacial Pain Subtypes

**DOI:** 10.3390/ijms26052260

**Published:** 2025-03-03

**Authors:** Weronika Jasinska, Yonatan Birenzweig, Yair Sharav, Doron J. Aframian, Andra Rettman, Aiham Hanut, Yariv Brotman, Yaron Haviv

**Affiliations:** 1Department of Life Sciences, Ben-Gurion University of the Negev, Beer-Sheva 84105, Israel; jasinska@post.bgu.ac.il; 2Hirszfeld Institute of Immunology and Experimental Therapy, Polish Academy of Sciences, 53-114 Wroclaw, Poland; 3Department of Oral Medicine, Sedation and Imaging, Hadassah Medical Center, Faculty of Dental Medicine, Hebrew University of Jerusalem, P.O. Box 12272, Jerusalem 91120, Israel; joni84@gmail.com (Y.B.); dorona@hadassah.org.il (D.J.A.); andrar@hadassah.org.il (A.R.); aihamh@hadassah.org.il (A.H.); yaronha@hadassah.org.il (Y.H.); 4School of Plant Sciences and Food Security, Tel Aviv University, Tel Aviv 6997801, Israel; brotman@tauex.tau.ac.il

**Keywords:** craniofacial pain, oral fluids metabolomics, chronic pain, biomarkers, metabolite profiling, burning mouth syndrome

## Abstract

Orofacial pain (OFP) includes chronic pain conditions categorized into musculoskeletal (MS), neurovascular (NV), and neuropathic (NP) pain types, encompassing temporomandibular disorders (TMD), migraines, trigeminal neuralgia (TN), post-traumatic neuropathies, and burning mouth syndrome (BMS). These conditions significantly affect quality of life; yet, their underlying metabolic disruptions remain inadequately explored. Salivary metabolomics provides a non-invasive method to investigate biochemical alterations associated with OFP subtypes. This study aimed to identify pain-specific salivary metabolites across chronic OFP types and examine their correlations with clinical characteristics. Saliva samples from 63 OFP patients (TMD, migraines, TN, post-traumatic neuropathies, BMS) and 37 pain-free controls were analyzed using liquid chromatography–mass spectrometry (LC-MS) targeting 28 metabolites linked to pain. Statistical analyses determined significant metabolite changes and associations with pain subtypes and patient characteristics. Among the 28 analyzed metabolites, 18 showed significant differences between OFP patients and controls. Key amino acids, including DL-glutamic acid, DL-aspartic acid, DL-citrulline, spermidine, and DL-ornithine, were significantly elevated in MS, NV, and NP pain types compared to controls. Additionally, DL-glutamine, DL-valine, and DL-phenylalanine were distinctively elevated in TMD and migraine patients. BMS displayed fewer alterations, with significantly lower levels of DL-proline, DL-tryptophan, DL-glutamic acid, DL-asparagine, and DL-aspartic acid compared to other pain types but elevated spermidine levels relative to controls. Salivary metabolomics revealed distinct metabolic alterations in OFP subtypes, providing insights into potential biomarkers for diagnosis and monitoring. These findings offer a foundation for personalized approaches in OFP management, although further research is required to validate and expand these results.

## 1. Introduction

Acute or chronic pain triggers the nervous system, leading to altered neural signaling, stress responses, and inflammation. It affects the hormonal balance, influencing the metabolism, cardiovascular function, and immune responses. Over time, persistent pain can disrupt sleep, emotional well-being, and cognitive functions, creating a cascade of physiological and psychological effects that touch virtually every system in the body. Epidemiological studies indicate that 10% or up to 25% of the general population report experiencing OFP, with 11% classified as chronic. Chronic OFP is associated with significant morbidity, psychological burden, and high utilization of healthcare resources [1].

Chronic OFP is broadly classified into three primary categories: neuropathic pain, neurovascular pain (e.g., migraines), and musculoskeletal pain (e.g., temporomandibular joint disorders, TM) [2]. Diagnostic standards for headaches and facial pain include the International Classification of Headache Disorders (ICHD) [3] and the International Classification of Orofacial Pain (ICOP) [4]. Tension-type headaches (TTH) and TMD are among the most commonly diagnosed headache and facial pain conditions. However, there is limited research exploring the metabolomics associated with these disorders. For instance, D’Andrea et al. [5] reported reduced plasma levels of tryptamine in chronic TTH patients, suggesting a role in the nociceptive processes mediated by serotonin receptors [5]. Similarly, Sanches et al. identified significant salivary metabolite alterations in women with TMD, including increased levels of isovalerate and acetoin and decreased levels of phenyl acetate and dimethylamine [6]. Other studies, such as Mantyselka et al. [7] and Livshits et al. [8], have demonstrated metabolic changes associated with chronic widespread pain, implicating roles in tissue repair and blood pressure regulation [7,8].

Neurovascular pain conditions such as chronic migraine (CM) and cluster headaches (CH) also exhibit distinct metabolic alterations. Dysregulation of the kynurenine pathway has been identified in CM and CH [9,10], and alterations in serotonergic metabolites have been reported in these conditions [11,12]. Additionally, elevated polyamine levels, as found by Aczel et al., may influence glutamate receptor function, contributing to pain perception [13,14].

Neuropathic pain conditions, including post-traumatic neuropathies, burning mouth syndrome (BMS), and trigeminal neuralgia (TN), have been associated with distinct salivary metabolite profiles, highlighting their potential as biomarkers for these disorders [15,16,17].

Metabolites, small molecules produced during metabolic processes such as energy generation and waste elimination, serve as key indicators of cellular and tissue-level biochemical activity. Alterations in metabolite levels can reflect disruptions in metabolic pathways associated with chronic pain conditions, offering valuable insights into the biological mechanisms underlying pain. Metabolomics, in particular, has been instrumental in uncovering the genetic basis of metabolic variation [18].

Saliva, which contains trace amounts of plasma, mirrors blood composition through passive and active transport as well as extracellular ultrafiltration [19]. It contains diverse genomic, transcriptomic, proteomic, microbiologic, and immunologic analytes, making it a valuable non-invasive diagnostic medium. The potential of saliva for early diagnosis, prognosis, and therapeutic monitoring is increasingly recognized [20].

The identification and quantification of metabolites in saliva present a promising approach for the early diagnosis and monitoring of treatment response in pain conditions. Encouraging findings in metabolomics studies of TMD and BMS underscore the potential of salivary metabolomics to advance the understanding and management of chronic primary headaches and OFP. Such advancements hold promise for personalized pain management and the development of targeted therapies.

This study aims to investigate the relationship between salivary metabolite levels and different types of orofacial pain, focusing on the correlation between pain characteristics and 28 specific metabolites previously identified in association with various pain conditions.

## 2. Results

Orofacial pain can be classified into three subcategories: musculoskeletal (MS), neuropathic (NP), and neurovascular (NV). The patients in this study were diagnosed with one of these subtypes, with 24 experiencing neuropathic pain, 16 diagnosed with neurovascular pain, and 22 classified as having musculoskeletal pain.

An additional group of 37 patients, who did not experience any orofacial pain, served as the control group. After metabolite extraction, 28 compounds were identified, with 7 showing a decrease in abundance in the control group compared to each of the subcategories and an increase for one metabolite for a control and decrease for the other types (Figure 1).

Further analysis revealed additional metabolites that differed between the control group and specific orofacial pain subgroups. A comparison between the control group and patients with musculoskeletal pain showed higher abundance in the control group for amino acids such as glutamic acid, proline, valine, ornithine, aspartate, urocanic acid, and asparagine. In contrast, theobromine, N-acetyl-phenylalanine, 4-hydroxybenzaldehyde, nicotinamide, and adenine showed decreased abundance (Figure 2 and Appendix A).

Another comparison was made between the control group and patients with neurovascular pain. Two out of three metabolites—theobromine and 4-hydroxybenzaldehyde—showed increased abundance in the control group, while urocanic acid showed decreased abundance (Figure 3 and Appendix A). These metabolites exhibited the same pattern of changes as observed in the musculoskeletal pain group. Data and *p*-values are provided in the Appendix A.

In the neuropathic pain group, the metabolite N-acetyl-phenylalanine showed lower abundance in the control group (Figure 4 and Appendix A), similar to the pattern observed in the comparison between the control and musculoskeletal pain groups. Data and *p*-values are provided in the Appendix A.

### Significant Metabolites to Pain (As Compared to Control) According to Pain Types

Patients classified into the three subcategories MS, NP, and NV were further diagnosed and divided based on the type of pain. Patients suffered from temporomandibular disorders (TMD), migraines (Mig), trigeminal neuralgia (TN), painful post-traumatic trigeminal neuropathy (PTN), burning mouth syndrome (BMS), and tension-type headache (TTH). Based on the analysis, there are 22 metabolites that showed a significant difference between the control and various types of pain (Table 1, Appendix A). Glutamine appeared to be the only metabolite that was abundantly different between the types of pain.

## 3. Discussion

Pain is a subjective experience, and objective signs are usually rare, especially in chronic pain conditions. The detection of objective findings, such as salivary metabolomics’ markers, could enhance the diagnosis, particularly in chronic pain cases with ambivalent symptoms.

Metabolomics, the comprehensive study of small molecules in biological samples, offers profound insights into an organism’s metabolic state, serving as a powerful tool for identifying disease biomarkers and tracking disease progression. Despite the immense potential of this field, many small molecules remain uncharacterized. However, existing research has highlighted metabolomic and proteomic dysregulation in various chronic diseases, including cancer, neurodegenerative disorders (e.g., Alzheimer’s and Parkinson’s diseases), pain syndromes, periodontal diseases, and other pathological conditions [19,21,22]. These findings underscore the need for further exploration of metabolomics to uncover the intricate biochemical pathways underpinning these conditions.

Both saliva and plasma are prominent biofluids in metabolomics research. Saliva, due to its non-invasive and easily accessible nature, presents a particularly attractive option [19]. While most compounds present in blood are also found in saliva, they appear at lower concentrations, as they transition into saliva via passive and active transport mechanisms and extracellular ultrafiltration. Despite these advantages, saliva has been underutilized in metabolic profiling compared to other biofluids like urine and plasma, as evidenced by the relatively limited body of research in this area [19]. Particularly, there is a paucity of data on salivary metabolites related to headache and OFP, highlighting a significant gap in the literature.

This study is innovative in its approach to exploring salivary metabolites specifically associated with chronic OFP across different pain subtypes. We analyzed a total of 28 metabolites selected for their roles in nociceptive signaling, neuroinflammation, and oxidative stress key pathways in chronic pain. These metabolites were chosen based on prior research [8,23,24,25], which identified their alterations in various chronic pain conditions, highlighting their potential as biomarkers for distinguishing orofacial pain subtypes. Their presence in saliva allows for non-invasive analysis, offering a practical approach to studying metabolic disruptions in pain mechanisms.

Of these, 18 metabolites (64.2%) exhibited significant changes in salivary levels in chronic OFP patients compared to controls. These included metabolites such as theobromine, N-acetylneuraminic acid, spermidine, DL-phenylalanine, DL-tryptophan, DL-isoleucine, DL-glutamic acid, DL-glutamine, urocanic acid, D-(+)-pyroglutamic acid, DL-asparagine, DL-valine, DL-aspartic acid, DL-citrulline, DL-ornithine, DL-proline, N-acetylphenylalanine, and uric acid. Among these, 12 metabolites demonstrated significant differences between pain groups and controls, with no significant gender differences observed. These included DL-glutamic acid, spermidine, DL-citrulline, D-pyroglutamic acid, DL-aspartic acid, DL-glutamine, DL-phenylalanine, DL-isoleucine, DL-tryptophan, DL-asparagine, DL-valine, and uric acid.

We assessed the relationship between these metabolites and various patient and pain characteristics. Significant correlations were observed across pain subgroups/types, while no associations were found with BMI, pain intensity measured by VAS, pain laterality (unilateral versus bilateral), nocturnal awakening due to pain, or pain duration.

Comparison with previous studies identified five key metabolites—DL-glutamic acid, DL-aspartic acid, DL-citrulline, spermidine, and DL-ornithine—that were either significantly correlated with OFP pain in plasma/serum studies or uniquely identified in our salivary metabolomics analysis [5,7,11,12,13,14,26,27,28]. In our study, DL-glutamic acid, DL-aspartic acid, DL-citrulline, and spermidine were highly significant across all pain subgroups (MS/NV/NP) compared to controls (*p* < 0.01), while DL-ornithine showed significance in MS and NV subgroups (*p* < 0.05).

These identified metabolite alterations align with established biochemical mechanisms underlying pain:

Glutamic and Aspartic Acids: Elevated levels are linked to hyperalgesia in neuropathic conditions like CRPS and are involved in nociceptive signaling via the blood–brain barrier [28].

Spermidine: Implicated in chronic migraine, it may regulate glutamate receptors, modulating pain responses [13,14].

Citrulline and Ornithine: Associated with nitric oxide pathways and tissue repair mechanisms, these metabolites have been linked to CM and CH [7,26,27].

Pyroglutamic Acid: As a marker of glutathione depletion, it indicates oxidative stress as a potential contributor to pain mechanisms [29].

Urocanic Acid: While not directly associated with pain, it may influence glutamate metabolism, a key player in nociception [30].

BMS represents a unique and complex condition with an unclear etiology. Unlike other chronic pain types, BMS patients showed fewer significant alterations in metabolite levels. Notably, five metabolites—DL-proline, DL-tryptophan, DL-glutamic acid, DL-asparagine, and DL-aspartic acid—were significantly lower in BMS patients compared to other pain types or controls, suggesting distinct metabolic dysregulation. Spermidine was the only metabolite significantly elevated in BMS patients compared to controls, reflecting its potential role in the condition’s unique pain mechanisms. Although limited research exists on BMS salivary metabolomics, evidence points to alterations in neuropeptides, cytokines, and hormones. Potential pathways include dysregulation of neurotrophin signaling and dopamine-mediated neural apoptosis [31]. The observed low metabolite levels may reflect these altered pathways, warranting further investigation.

The diagnostic implications of these findings are profound. Saliva, as a non-invasive and accessible biofluid, is uniquely positioned to facilitate early detection, differential diagnosis, and monitoring of therapeutic responses in OFP. For instance, the ability to differentiate between TMD and neuropathic conditions using metabolomic signatures can enable more targeted interventions, reducing diagnostic uncertainty and enhancing patient outcomes. Additionally, the identification of metabolites like spermidine as potential therapeutic targets highlights the significance of metabolite sciences in advancing pain management strategies.

Limitations: The study does not address the bidirectional relationship between metabolites and pain. Pain can induce metabolic changes that affect energy pathways, inflammation, and oxidative stress, while altered metabolite levels can influence pain sensitivity and persistence. This dynamic interaction underscores the importance of adopting integrative approaches that combine metabolomics with other diagnostic and therapeutic methods, providing a more comprehensive understanding of pain mechanisms. Limited patient availability and time constraints may impact the statistical power, so increasing the patient cohort and including technical controls would be beneficial.

## 4. Materials and Methods

### 4.1. Criteria and Collection of Medical Records

All data were fully anonymized, and informed consent was waived in accordance with the Ethics Committee’s instructions. Medical records of 100 participants were reviewed, among which, 63 individuals attended the Orofacial Pain Clinic at the Hebrew University-Hadassah Faculty of Dental Medicine between 2017 and 2018, while the remaining 37 participants served as an age-matched pain-free control group. The inclusion criteria were age ≥ 18 years and diagnosis of chronic orofacial pain persisting for at least three months.

Exclusion criteria included the following: refusal or inability to provide consent; medical conditions affecting salivary gland function, such as autoimmune diseases (e.g., Sjögren’s syndrome, rheumatoid arthritis); a history of head and neck cancer treated with radiotherapy, chemotherapy, or biologically targeted therapies; graft-versus-host disease; fibromyalgia; and a salivary flow rate of less than 0.2 mL/10 min.

### 4.2. Orofacial Pain Diagnosis

Participants with orofacial pain were classified into three diagnostic groups—musculoskeletal, neurovascular (migrainous), and neuropathic—based on etiology and pain characteristics, as outlined in Sharav and Benoliel’s reference book [2]. Each category was further subdivided into specific diagnoses, providing a structured and clinically relevant classification aligned with established diagnostic frameworks.

#### 4.2.1. Musculoskeletal Group

Included participants with pain related to temporomandibular disorders (TMD), including masticatory muscle pain (MMP), temporomandibular joint (TMJ) pain, or combined muscle and joint pain. Diagnoses were established according to the Diagnostic Criteria for Temporomandibular Disorders (DC/TMD) [32].

#### 4.2.2. Neurovascular Group

Although chronic tension-type headache is primarily categorized as a musculoskeletal disorder, participants presenting additional neurovascular symptoms or underlying pathophysiology were included. Diagnoses of migraine and tension-type headache followed the International Classification of Headache Disorders, 3rd Edition (ICHD-3) [3]. Neurovascular orofacial pain (NVOP) was diagnosed for facial pain exhibiting migrainous features within the second and/or third divisions of the trigeminal nerve [33].

#### 4.2.3. Neuropathic Group

Included participants with trigeminal neuralgia (TN) [34], painful post-traumatic trigeminal neuropathy (PTN) [35], persistent idiopathic facial pain (neuropathic unspecified pain) [36], and burning mouth syndrome (BMS) [37].

A detailed extra oral examination included cranial nerve assessment and palpation of the masticatory apparatus [38,39]. An intraoral examination was conducted to rule out dental, periodontal, and mucosal pathology. Brain and brainstem imaging were performed for participants with TN to exclude intracranial pathology.

### 4.3. Collection of Medical Records

Medical records were reviewed for demographic information (e.g., gender, age, BMI), medication use, and relevant medical history. Pain characteristics, including onset (duration in months), intensity, and quality, were recorded during the intake process routinely employed in the clinic. Pain intensity was assessed using a verbal pain scale (VPS) on the day of saliva collection, where 0 represented no pain and 10 denoted the worst imaginable pain.

### 4.4. Saliva Collection

Unstimulated saliva samples were collected over a 10 min period into pre-calibrated tubes, adhering to established protocols [31,40]. Collections took place between 9:00 a.m. and 12:00 p.m. to minimize diurnal variability. Participants were instructed to abstain from physical activity, eating, drinking, and oral hygiene (e.g., brushing teeth) for at least two hours prior to collection. Patients were also required to refrain from taking any medications, including sialagogues, before the procedure. Prior to sample collection, participants rested for 10 min in a seated position in a quiet room and were instructed not to speak or leave the room until the procedure was complete. Immediately after collection, saliva samples were stored at −80 °C for preservation until further processing, in Prof. Yariv Brotman’s laboratory at the Department of Life Science, Ben-Gurion University of the Negev, Beersheba, Israel.

### 4.5. Sample Preparation

The saliva samples were thawed and centrifuged at 3500× *g* for 10 min at 2 °C to re-move insoluble materials, including cell debris and food remnants. The supernatant was aliquoted into polypropylene tubes and stored at −80 °C until further analysis. Samples were processed systematically, with two OFP samples from the same group followed by a gender- and BMI-matched control sample. The samples were analyzed in the following sequence: 2 CFP samples from the same groups, followed by a gender and BMI-matched control sample. The saliva samples were prepared and analyzed by liquid chromatography–mass spectrometry (LC-MS) for metabolite identification and quantification. The extraction of polar metabolites was performed as previously described [41]. To each sample, 1 mL of a pre-cooled mixture of methanol/methyl-tert-butyl-ether/water (1:3:1) was added, followed by vortexing. The samples were shaken for 10 min and incubated in an ice-cooled ultra-sonication bath for 10 min. Then, 500 μL of UPLC-grade methanol/water (1:3) was added to the homogenate, which was vortexed and centrifuged for 5 min at 4 °C, resulting in phase separation with polar and semi-polar metabolites in the lower aqueous phase. An equal volume of 300 μL from the lower phase was transferred to a fresh Eppendorf tube, dried, and re-suspended in 80% (*v*/*v*) methanol and 20% (*v*/*v*) water before injection into the LC-MS system (Waters Acquity UPLC system, coupled to an Exactive mass spectrometer by Thermo Fisher Scientific, Waltham, MA, USA). Instrumental settings were previously described [42]. Data extraction and analysis were conducted using Xcalibur™ Software 4.5 (Thermo Fisher Scientific) and Compound Discoverer 3.3 (Thermo Fisher Scientific).

### 4.6. Statistical Analysis and Data Visualization

To examine the differences in salivary metabolites levels for nominal and categorical background variables, a *t*-test and one-way analysis of variance were performed. The differences between salivary metabolites and specific background variables were analyzed for each diagnosis separately by using the non-parametric Mann–Whitney test. Figures and *p*-values were obtained after calculations from the website https://new.metaboanalyst.ca/ (accessed on 12 May 2024) and software GraphPad Prism 9.0.0.

## Figures and Tables

**Figure 1 ijms-26-02260-f001:**
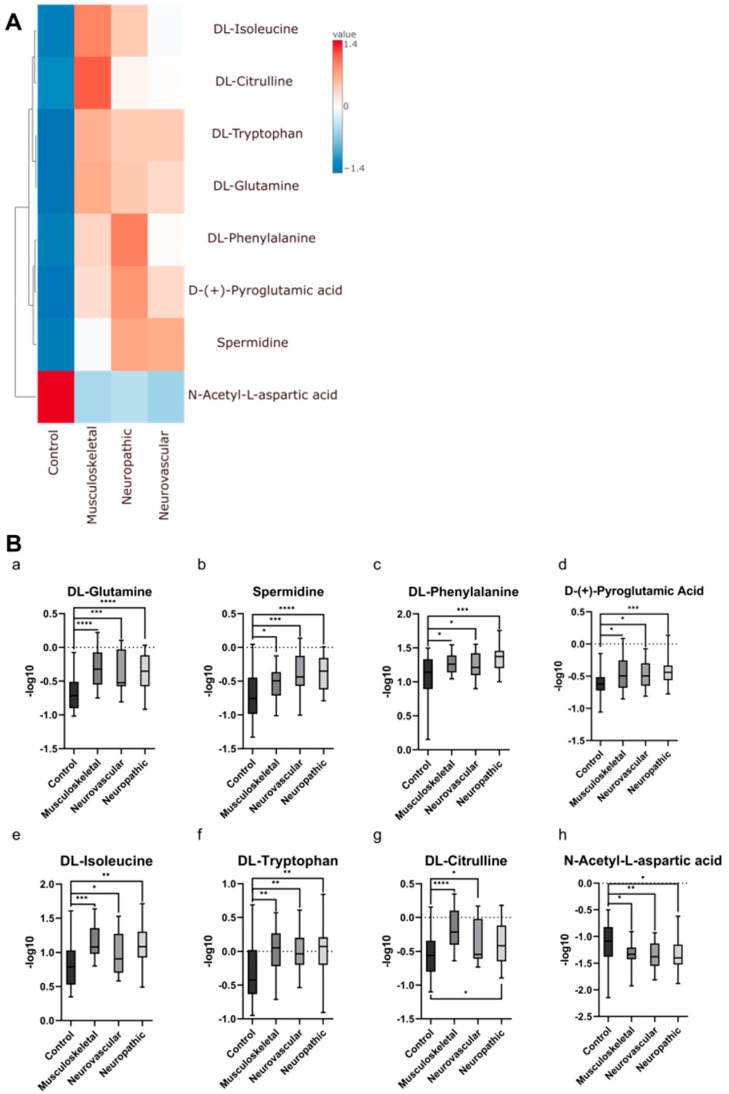
Heatmap (**A**) and boxplots (**B**) presenting eight metabolites ((**a**) DL-glutamine, (**b**) spermidine, (**c**) DL-phenylalanine, (**d**) D-(+)-pyroglutamic acid, (**e**) DL-isoleucine, (**f**) DL-tryptophan, (**g**) DL-citrulline, (**h**) N-acetyl-L-aspartic acid) showing a significant change in abundance for a control group versus the pain subcategories. Data and *p*-values are provided in the Appendix A. * *p*-value < 0.05; ** *p*-value < 0.005; *** *p*-value < 0.001; **** *p*-value < 0.0001.

**Figure 2 ijms-26-02260-f002:**
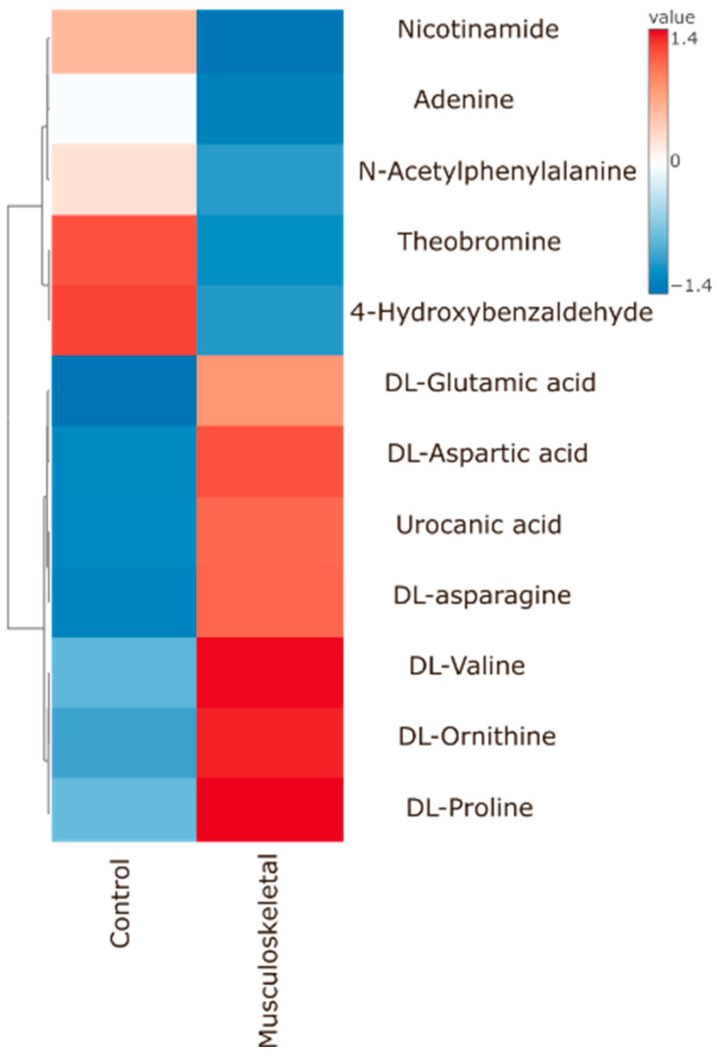
Heatmap presents 12 metabolites that were found to be significantly different between control group and patients diagnosed with musculoskeletal pain. Data and *p*-values are provided in the Appendix A.

**Figure 3 ijms-26-02260-f003:**
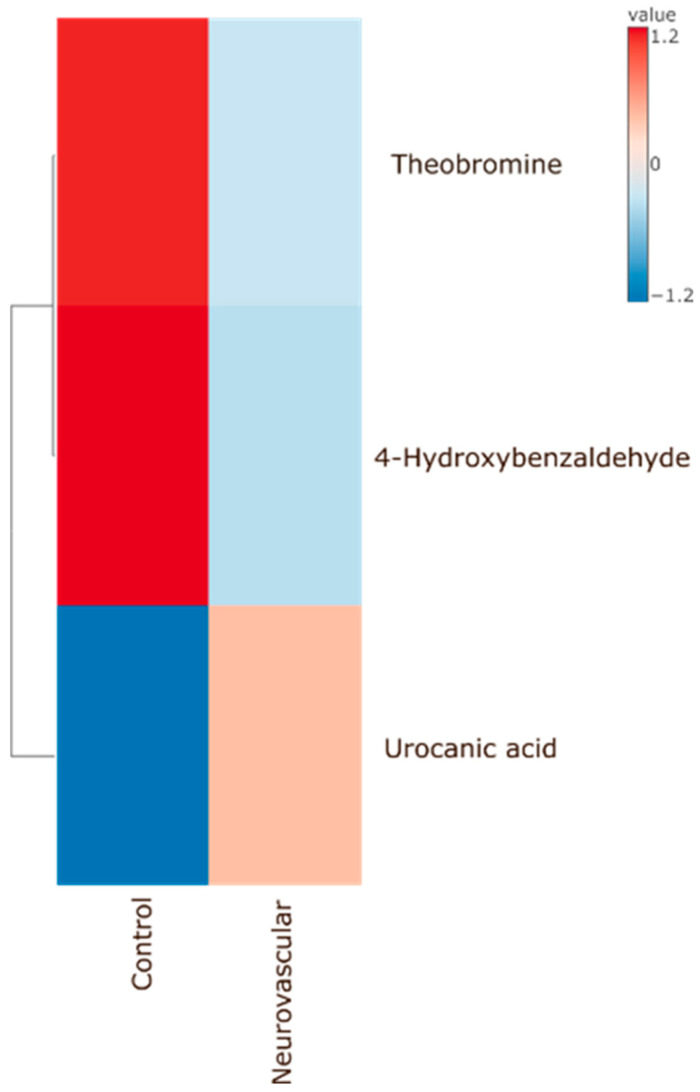
Heatmap presents three metabolites that were found significantly different between control group and patients diagnosed with neurovascular pain.

**Figure 4 ijms-26-02260-f004:**
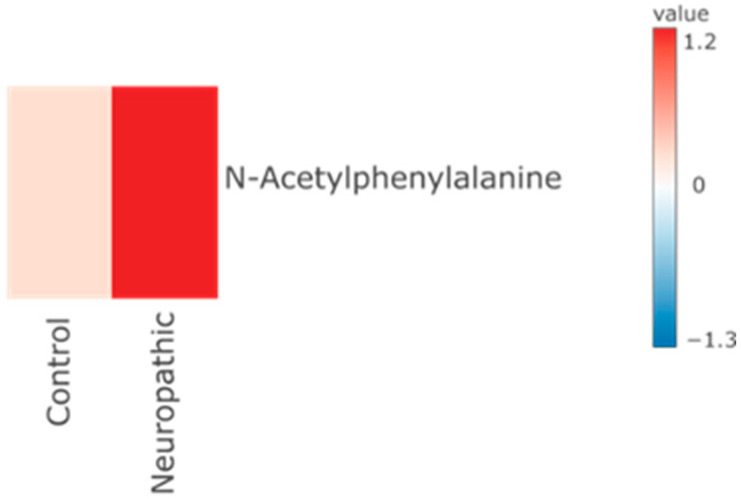
Heatmap presents three metabolites that were found significantly different between control group and patients diagnosed with neuropathic pain.

**Table 1 ijms-26-02260-t001:** Table includes metabolites whose intensities significantly differ between control group and types of pain. Mann–Whitney test was used to calculate the *p*-values.

*p*-Values						
**control vs. Mig/TMD**						
4-Hydroxybenzaldehyde	0.0237	0.0213				
DL-Valine	0.0324	0.0009				
N-Acetylphenylalanine	0.0389	0.0155				
Theobromine	0.0036	0.0001				
**control vs. TMD**						
Adenine	0.0182					
DL-Ornithine	0.0013					
L-Histidine	0.0428					
Nicotinamide	0.0262					
**control vs. PTN/TN/TTH**						
D-(+)-Pyroglutamic Acid	0.0068	0.0084	0.0008			
DL-Glutamic acid	0.0326	0.0249	0.013			
**control vs. TTH**						
DL-Asparagine	0.0008					
**control vs. PTN/TMD/TN/TTH**						
DL-Aspartic acid	0.0326	0.0082	0.0227	0.0021		
DL-Citrulline	0.0268	<0.0001	0.0189	0.0246		
DL-Isoleucine	0.0145	0.0002	0.0068	0.0027		
**control vs. BMS/Mig/PTN/TMD/TN/TTH**						
DL-Glutamine	0.0434	0.0026	0.0015	<0.0001	0.0003	0.0009
**control vs. PTN/TMD/TN**						
DL-Phenylalanine	0.0045	0.0354	0.0048			
**control vs. BMS/TMD**						
DL-Proline	0.0329	0.0002				
**control vs. Mig/PTN/TMD/TN/TTH**						
DL-Tryptophan	0.0061	0.0156	0.0044	0.0128	0.0396	
**control vs. TN**						
DL-Tyrosine	0.0063					
**control vs. BMS/Mig/TMD/TN/TTH**						
N-Acetyl-L-aspartic acid	0.0222	0.0408	0.0249	0.0324	0.0246	
**control vs. BMS/Mig/PTN/TN**						
Spermidine	0.0116	0.0016	0.0008	0.0156		
**control vs. Mig/TMD/TTH**						
Urocanic acid	0.0172	0.0002	0.0474			

## Data Availability

Data is contained within the article and Appendix A.

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
