# Peer review of "Salivary Metabolomics as a Diagnostic Tool: Distinct Metabolic Profiles Across Orofacial Pain Subtypes"

_ijms, 2025, doi:10.3390/ijms26052260_

Round 1

Reviewer 1 Report

Comments and Suggestions for Authors

The article proposes a saliva metabolomic analysis as a diagnostic tool for chronic orofacial pain (OFP), divided into musculoskeletal (MS), neurovascular (NV) and neuropathic (NP) subtypes. The study aims to identify specific metabolites for each subtype and evaluate their correlation with clinical characteristics of patients.
The work is well structured and follows a clear logical flow: introduction, materials and methods, results, discussion and conclusion.
However, the justification for the selection of 28 analyzed metabolites should be better explained with more solid references to the literature. Furthermore, a detailed explanation on the reason for the subdivision of patients into diagnostic groups is not provided, an aspect that could influence the interpretation of the data.
The text is clear and well organized, but there are some repetitions in the discussion (e.g. the role of some metabolites is described several times).
The use of salivary metabolomics for orofacial pain is an emerging field and the work fits into an innovative research area. However, the literature review should be expanded to include more recent and relevant studies that have applied similar approaches in other chronic pain conditions.
If confirmed by future studies, the results could have significant diagnostic and therapeutic implications for chronic orofacial pain. However, the relatively small number of participants and the lack of independent validation limit the immediate impact of the study.
The references include recent and relevant references, but some key studies on metabolomics in chronic pain conditions are not cited. It would be useful to integrate additional references to provide a more complete picture of the existing literature.
The figures are clear and well structured.
Suggestions for the review
• Clarify the choice of the 28 metabolites analyzed, providing solid references.
• Expand the literature review to include recent studies on metabolomics in chronic pain.
• Improve the description of patient selection criteria and better justify their division into subgroups.
• Strengthen the statistical rigor of the analyses and include technical controls.

Reviewer 2 Report

Comments and Suggestions for Authors

The manuscript entitled “Salivary Metabolomics as a Diagnostic Tool: Distinct Metabolic Profiles across Orofacial Pain Subtypes” by Jasinska and collaborators, which aimed to identify pain-specific salivary metabolites across chronic OFP types and examine their correlations with clinical characteristics, shows a very important theme about orofacial pain. The authors are to be congratulated for their excellent work, an innovative and promising approach to this problem that is often unknown to a portion of the population and for this reason there is a delay in treatment. I have very few considerations that follow...

Abstract

Lines 26-28: Metabolites could be replaced by amino acids.

Keywords

Lines 36-37: Replace the keywords orofacial pain and salivary metabolomics with others that are not part of the title.

Introduction

Line 77: Add final punctuation.

Results

Lines 96-98: Standardize the quantity either in words or numbers.

Lines 99-100: Same as above

Lines 104: Improve the resolution of the Figure, the texts are incomprehensible.

Line 110: They can be called amino acids.

Lines 113, 121, 129, 141: There are no Figures in the supplementary material, only Tables.

Lines 196 and 198: Missing final punctuation

Materials and methods

Line 238: Same age, but what age? Add.

Line 300: The experimental conditions should be added to the methodology so that the reader reading this work will need to look for the cited reference because the complete information is not available here.
